# An ultraviolet-driven rescue pathway for oxidative stress to eye lens protein human gamma-D crystallin
Jake A. Hill [1,2], Yvonne Nyathi[3], Sam Horrell [4], David von Stetten [5], Danny Axford [4], Robin L. Owen [4], Godfrey S. Beddard[2,6], Arwen R. Pearson [7], Helen M. Ginn [7,8,9] & Briony A. Yorke [2,9]

Human gamma-D crystallin (HGD) is a major constituent of the eye lens. Aggregation of HGD contributes to cataract formation, the leading cause of blindness worldwide. It is unique in its longevity, maintaining its folded and soluble state for 50-60 years. One outstanding question is the structural basis of this longevity despite oxidative aging and environmental stressors including ultraviolet radiation (UV). Here we present crystallographic structures evidencing a UV-induced crystallin redox switch mechanism. The room-temperature serial synchrotron crystallographic (SSX) structure of freshly prepared crystallin mutant (R36S) shows no post-translational modifications. After aging for nine months in the absence of light, a thiol-adduct (dithiothreitol) modifying surface cysteines is observed by low-dose SSX. This is shown to be UV-labile in an acutely light-exposed structure. This suggests a mechanism by which a major source of crystallin damage, UV, may also act as a rescuing factor in a finely balanced redox system.

Human gamma-D crystallin (HGD) is a soluble, monomeric protein expressed in the eye lens during fetal development. The high expression levels contribute to the refractive index within the lens which focuses visible light upon the retina. HGD is thought to provide some protection for the retina against UV due to its relatively high abundance of aromatic amino acids[1]. It has been suggested that HGD may also perform an oxidoreductase function helping to reduce oxidized proteins[2]. During the maturation of the lens, epithelial cells differentiate to form fiber cells in which cellular machinery is broken down to aid transparency. However, this halts further expression of HGD. This means HGD must remain correctly folded and soluble for the entire human lifespan in order to maintain visual acuity. Lens fiber cells contain the structural proteins beta and gamma crystallins and the chaperone protein alpha-crystallin, which is known to sequester and repair beta and gamma crystallins[3,4]. Aging, injury, systemic and genetic disease, exposure to radiation, toxins and environmental pollutants are all known to contribute to HGD degradation and in turn cataractogenesis[5]. Cataract is

the major cause of visual impairment worldwide (WHO) and is characterized by an aberrant decline in the ocular lens transparency. A variety of post-translational modifications have been detected in HGD purified from cataractous lenses and in the cellular environment, indicating that the pathways of UV damage, rescue and aggregation are numerous and complex[5]. Here we focus on damage rescue mechanisms in HGD related to UV radiation.

During everyday life, the lens is unavoidably exposed to UV radiation from sunlight. This results in both primary photodamage to aromatic groups and secondary damage to the protein through the formation of reactive oxygen species (ROS) which contribute to protein oxidation and negatively impacts stability. The role of high concentrations of the reducing agent glutathione (GSH) in the lens has been studied in great depth and GSH has been shown to successfully reduce the rate of secondary photodamage by acting as an ROS scavenger[6-10]. GSH is synthesized in the lens epithelial cells[11] and may prevent aggregation caused by the formation of

¹School of Chemistry and Biosciences, University of Bradford, Richmond Road, Bradford BD7 1DP, United Kingdom. ²School of Chemistry, University of Leeds, Woodhouse Lane, Leeds LS2 9JT, United Kingdom. ³Faculty of Biological Sciences, University of Leeds, Woodhouse Lane, Leeds LS2 9JT, United Kingdom. ⁴Diamond Light Source Ltd, Harwell Science and Innovation Campus, Didcot OX11 0DE, United Kingdom. ⁵European Molecular Biology Laboratory, Notkestraße 85, 22607 Hamburg, Germany. ⁶School of Chemistry, University of Edinburgh, David Brewster Road, Edinburgh EH9 3FJ, United Kingdom. ⁷HARBOR, Institute for Nanostructure and Solid State Physics, Hamburg 22761, Germany. ⁸Center for Free-Electron Laser Science, CFEL, Deutsches Elektronen-Synchrotron DESY, Notkestr. 85, 22607 Hamburg, Germany. ⁹These authors jointly supervised this work: Helen M. Ginn, Briony A. Yorke. ✉e-mail: crystallin@hginn.co.uk; b.a.yorke@leeds.ac.uk

intermolecular disulfide crosslinks by reversing the oxidation of surface thiol groups[12–15]. It has also been shown to suppress copper-induced aggregation[10]. Upon aging, the ratio of reduced GSH to the oxidized form GSSG decreases, due to disruption to synthesis and diffusion of GSH from the epithelial layer into the lens[16,17]. The depletion of the free GSH concentration and increased levels of protein S-glutathionylation and cysteine oxidation is observed in cataractous lenses[18].

The lens plays a role in protecting the retina from UV-induced damage. This is supported by the presence of UV-filtering molecules 3-hydroxykynurenine, kynurenine, and 3-hydroxykynurenine glucoside and the presence of a cluster of four highly conserved tryptophan residues (W42 and W68 in the N-terminal domain, W130 and W156 in the C-terminal domain) in HGD. While the presence of aromatic amino acids in HGD may intuitively be assumed to contribute to photodamage, these four tryptophan residues are likely to instead provide protection against UV-induced aggregation[19] possibly by quenching of the excited state by the neighboring tryptophan residue followed by electron transfer to backbone amides[20].

The study of HGD therefore spawns two conceptually opposite branches of research: how does it remain so stable in the first place, and what mechanisms eventually lead to its aggregation? In this paper, we concentrate on the mechanisms that maintain crystallin stability rather than the pathways that lead to disease.

In order to observe structural changes in HGD, we take advantage of an arginine-to-serine point mutation, R36S, first identified in congenital cataracts ($HGD_m$)[21]. This mutation encourages rapid crystallization due to changes in surface charge, leading to an increase in favorable crystal contacts[22]. Here, we present crystal structures of freshly prepared $HGD_m$ (fresh dataset), $HGD_m$ aged for 9 months in dark conditions (aged dataset), and similarly aged $HGD_m$ during exposure to UV (light dataset). Crystals

were prepared with 20 mM dithiothreitol (DTT), a sulfur-containing reducing agent with a similar function to that which GSH is expected to perform in vivo.

Although the fresh structure shows no significant binding of DTT, after aging of the crystals, DTT is seen to form disulfide bonds to two of the surface cysteine residues, C41 and C108 with strongly ordered density. We show that UV exposure is then able to disrupt the disulfide bond, and partially restore the conformation to that of fresh crystallin. We also characterize the subtle but significant changes to the crystallin structure accompanying DTT binding and UV-induced disulfide cleavage. With these structures we propose a mechanism of replenishment of the crystallin protein by UV-triggered recycling of the reducing agent.

## Results and Discussion
### Aging of crystals in the presence of DTT

The effect of oxidation of $HGD_m$ was investigated by incubating a slurry of microcrystals at room temperature in the absence of light for nine months. Structures from crystals before (Supplementary Data 1) and after (Supplementary Data 2) this process were compared for each of the chains (A, X) in the asymmetric unit (Fig. 1A, B). After aging, DTT adduct formation is observed at 100% occupancy at C41 and C108 and oxidation at C110 to sulfenic acid. Difference map peaks surrounding DTT in Fo-Fo maps of aged relative to fresh datasets were calculated as 10.1 r.m.s.d. (A-C41), 8.4 r.m.s.d. (A-C108), 9.9 r.m.s.d. (X-C41) and 10.1 r.m.s.d. (X-C108) (Fig. 1C–F) and are also clear in the 2mFo-dFc maps (Fig. 1G–J). The crystallographic data and refinement statistics are summarized in Table 1.

Although DTT is used in the preparation of many published HGD crystal structures[23–26], upon visual inspection of these, adduct formation was not observed in the electron density. However, these structures stem from

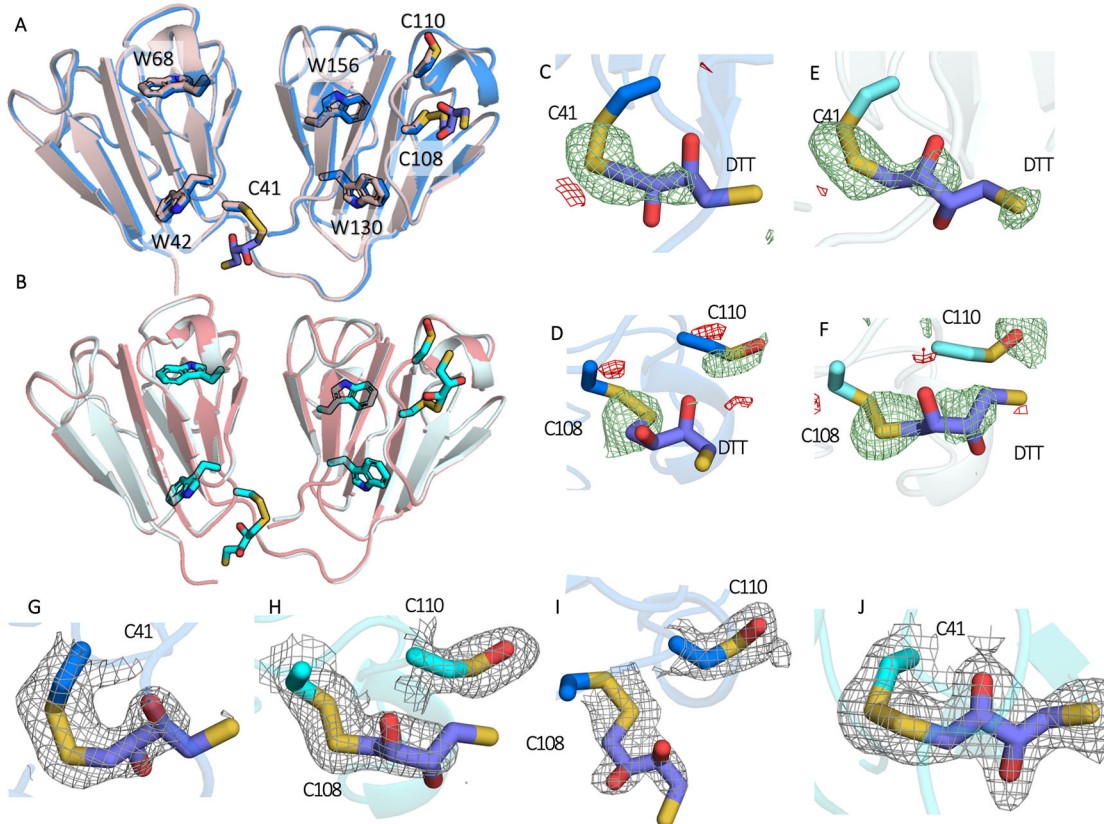

**Fig. 1 | Structural evidence for the formation of disulfide adducts between cysteine and DTT upon aging of HGD crystals. A** Chain A of the fresh structure (pink) overlaid with the aged structure (blue). **B** Chain X of the fresh structure (salmon) overlaid with the aged structure (cyan). Fo-Fo difference maps contoured at 3 σ calculated by subtracting the electron density of the fresh dataset from aged dataset centred on (**C**) chain A C41, (**D**) chain A C108, (**E**) chain X C41 and (**F**) chain X C108. 2mFo-dFc maps of the aged dataset contoured at 1 σ centred on (**G**) chain A C41, **H** chain X C41, (**I**) chain A C108 and (**J**) chain X C108.

**Table 1 | Crystallographic data collection and refinement statistics, values in parentheses are for the highest-resolution shell**

| | R36S HGD Fresh | R36S HGD Aged | R36S HGD Light (5 ms) |
|---|---|---|---|
| PDB | 8Q3L | 8BD0 | 8BPI |
| Wavelength (Å) | 0.99 | 0.98 | 0.98 |
| Resolution range (Å) | 64.3–2.10 (2.14–2.10) | 101.13–2.00 (2.07–2.00) | 101.00–2.00 (2.07–2.00) |
| Space group | P $2_1 2_1 2_1$ | P $2_1 2_1 2_1$ | P $2_1 2_1 2_1$ |
| Unit cell | | | |
| a, b, c (Å) | 53.9, 84.1, 101.5 | 53.9, 83.7, 101 | 53.9, 83.7, 101 |
| $\alpha, \beta, \gamma$ (°) | 90, 90, 90 | 90, 90, 90 | 90, 90, 90 |
| No. indexed images | 11,184 | 6192 | 7141 |
| Unique reflections | 31,571(1484) | 31,629 (3090) | 31,633(3090) |
| Multiplicity | 26.7 (17.24) | 50.3 (34.70) | 70.2 (48.30) |
| Completeness (%) | 99.90 (99.70) | 99.99 (100) | 100 (100) |
| Mean I/sigma(I) | 3.87 (0.95) | 2.71 (0.60) | 3.19 (0.68) |
| Wilson B-factor (Å²) | 18.7 | 26.8 | 28.0 |
| Rsplit | 0.19 (1.25) | 0.27 (1.29) | 0.23 (1.36) |
| CC$\frac{1}{2}$ | 0.97 (0.49) | 1.0 (0.19) | 0.95 (0.14) |
| CC* | 0.99 (0.81) | 0.98 (0.57) | 0.98 (0.50) |
| Reflections used in refinement | 25,669 | 25,054 | 27,045 |
| Reflections used for R-free | 1321 | 1718 | 1746 |
| Rwork (%) | 20.38 | 20.44 | 20.25 |
| Rfree (%) | 25.48 | 24.97 | 24.02 |
| Number of non-H atoms | 3011 | 3018 | 3012 |
| macromolecules | 2937 | 2905 | 2932 |
| ligands | 0 | 32 | 0 |
| solvent | 74 | 81 | 80 |
| RMS bonds (Å) | 0.007 | 0.008 | 0.01 |
| RMS angles (°) | 1.43 | 1.49 | 1.75 |
| Ramachandran | | | |
| favored (%) | 99.00 | 98.49 | 99.08 |
| allowed (%) | 1.00 | 1.51 | 0.92 |
| outliers (%) | 0 | 0 | 0 |
| Rotamer outliers (%) | 1.00 | 2.26 | 1.60 |
| Clashscore | 5.00 | 4.93 | 4.40 |
| Average B-factor (Å²) | 31.4 | 37.8 | 39.3 |
| - macromolecules | 31.4 | 37.5 | 39.4 |
| - ligands | | 74.7 | |
| - solvent | 36.5 | 36.1 | 37.6 |

single-crystal diffraction experiments. It is likely that DTT is rapidly lost due to the vulnerability of the disulfide bond to radiation damage. SSX takes single snapshots of many crystals with limited X-ray exposure per crystal, hence there is much-reduced dose and radiation damage[27]. In our structures, the fresh HGD$_m$ crystals incurred an average diffraction-weighted dose (DWD)[28] of 0.19 MGy after 20 ms X-ray exposure[29]. In contrast, the aged structure incurred a radiation dose of just 0.02 MGy. To control for the possibility of radiation damage-induced loss of DTT in the fresh structure, a series of 20 consecutive 5 ms exposures of each aged crystal was carried out (burst series) to accumulate a DWD of 0.31 MGy. The structure from the final timepoint showed no loss of DTT. Fo-Fo difference maps calculated between the first and last timepoint revealed no peaks above 3.0 r.m.s.d., confirming that X-ray-induced damage was not responsible for the absence of Cys-DTT in the fresh HGD$_m$ structure.

In addition to these covalent modifications, the protein conformation is subtly different, with an overall r.m.s.d. of 0.23 Å for chain A and 0.21 Å for chain X. However, this is dwarfed by the 0.67 Å r.m.s.d. between chains A and X in the aged structure and 0.62 Å in the fresh structure. It was important to determine if this conformational change upon binding was significant with respect to the changing conditions, rather than natural exploration of local conformations at the ground state.

## Effect of UV radiation on the aged crystals

Data were collected to investigate the effect of 5 ms UV radiation on aged crystals (Table 1 (Supplementary Data 3). To determine whether the structural differences between the fresh, aged, and light samples were larger than the difference between independent structures of each (i.e. fresh vs. fresh, etc.) We took advantage of the oversampling of the SSX experiment to split each SSX dataset into five sub-datasets that each contained sufficient images for stable processing of refinement. As each sub-dataset contained a different crystal population, they can be regarded as truly independent. Fully automated refinement from the same starting structure was carried out for each of the 15 split datasets, and resulting structures were passed through RoPE[30], to look for condition-specific separation in conformational space through analysis of atomic coordinates. Separation occurred in the atomic coordinate-derived spaces between the three conditions and for each chain (Fig. 2A, B). Inspection of the torsion angle heat maps (Fig. 2C, D) showed that the torsion angle changes occurring in chain A and chain X bear no relation to each other. However, in atomic coordinate space, the chains had more similar profiles to one another. Due to the stark separation, the changing condition aligned with the first principal component of the singular value decomposition (SVD) analysis. This principal component, which corresponds to a series of atomic coordinate motions, was converted into a matrix showing the change in interatomic distances. First, the difference in distance plots between chain A and chain X shows an expansion of the core of the N-terminal domain (NTD) and a contraction in that of the C-terminal domain (CTD) in chain X relative to A (Fig. 2E). Aging of chain A leads to an expansion of the NTD and a contraction in the CTD (Fig. 2F). Aging of chain X leads to a contraction in both domains, and a notable reduction in the distance between the two (Fig. 2G).

UV-lability of the Cys-DTT disulfide was tested by exposing similarly aged crystals to a UV LED at 285 nm during X-ray data collection. In the first 5 ms dataset, DTT adduct cleavage is visible at C41 and C108 in both chains (Fig. 3A–D), with oxidation to sulfenic acid observed at C41 in both chains. Difference map peaks around DTT when subtracting observed amplitudes of aged from light datasets are -6.7 r.m.s.d. (A-C41), -5.8 r.m.s.d. (A-C108), -5.4 r.m.s.d. (X-C41) and -9.4 r.m.s.d. (X-C108) (Fig. 3A–D). The solvent accessibility and pKa of the three surface cysteine residues were calculated from the fresh structure using PropKa3[31] and PyPka[32] (Table 2). Both methods predicted the pKa of C110 to be lower than that of C41 and C108. This lower pKa would be consistent with stabilization of sulfenic acid, as observed in the crystal structure. However, it should be noted that the physiological pH of lens fiber cells is 6.8, while the pH of the crystallization buffer is 8.0.

The exact pathway or pathways of UV-induced disulfide cleavage is worthy of discussion. Direct photocleavage to produce thiyl radicals, while possible, is unlikely to be the major contribution to the observed structural changes after 285 nm illumination. The $\lambda_{max}$ absorption of disulfide bonds is dependent on the C-S-S-C dihedral angles as shown in Table 3 with extinction coefficients on the order of a few hundred M$^{-1}$ cm$^{-1}$ [33]. Each sulfur atom has two lone pair orbitals, one $s$-type and one $p$-type. The minimum energy configuration of the C-S-S-C dihedral is around 90° where the lone pair $p$-orbitals are orthogonal to one another. This corresponds to a peak in the molar absorptivity around 260 nm. This has a minimum oxidation potential. Divergence from the optimal dihedral also results in increased overlap between the sulfur $p$-orbital lone pair and the adjacent

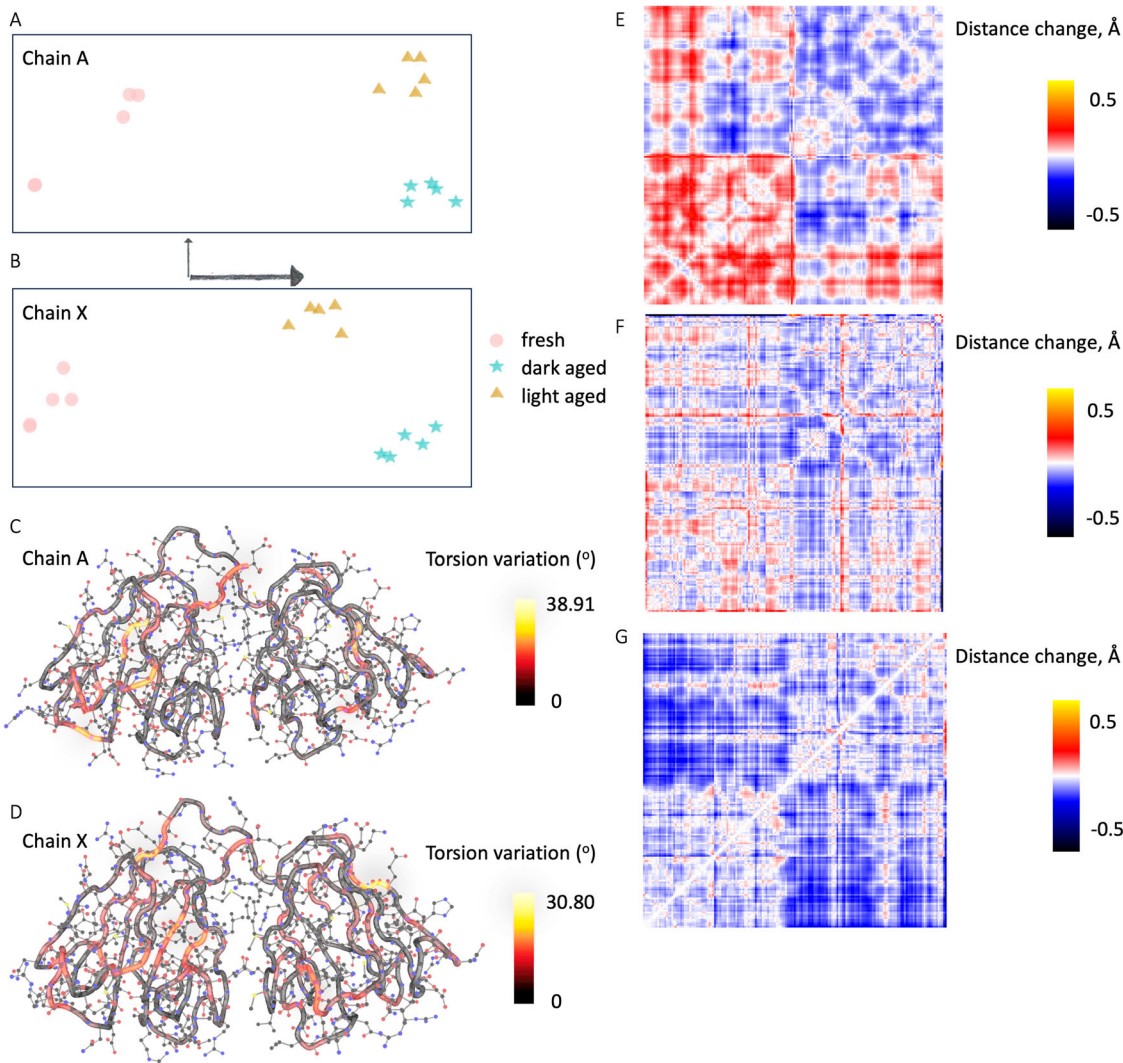

**Fig. 2 | Analysis of torsion angle variance between fresh, aged and UV-exposed HGD.** RoPE space[30] calculated using atom positions for each of the batched datasets for (**A**) chain A and (**B**) chain X. Axes indicate relative length of first two principal components. Heat maps of main chain torsion angle variance per residue for horizontal axes of RoPE spaces for (**C**) chain A and (**D**) chain X. **E–G** Atom-to-atom distance difference plots showing global changes exhibited by the first principal component axis of RoPE space for Cα atoms. N-terminus in bottom left corner, C-terminus in top right corner. Plots drawn for (**E**) chain X relative to chain A, **F** aged structure relative to the fresh structure, chain A and (**G**) aged structure relative to the fresh structure, chain X.

carbonyl non-bonding $\pi^*$ orbital, decreasing the likelihood of homolytic disulfide bond cleavage. Sulfur K-edge X-ray absorption spectroscopy shows that at 267 nm, UV-induced cleavage of the disulfide occurs via an excited state, identified from the $1s \rightarrow \pi^*$ transition[34]. This is accompanied by C-S bond cleavage identified from the $1s \rightarrow \sigma^*$ transition, leading to the formation of perthiyl radicals. At longer wavelengths the C-S cleavage is thought to dominate. As there is no evidence in the structure of loss of terminal sulfur density, we can assume that UV-induced direct disulfide cleavage is not the dominant pathway.

**Mechanism of UV-interaction with aged crystals**

The cluster of four conserved tryptophan residues in the hydrophobic core of the protein have been previously implicated in HGD stability[2]. The extinction coefficient of tryptophan in hydrophobic environments is 4694 $M^{-1} cm^{-1}$ at 285 nm[35], much higher than that of the C-S-S-C disulfide bond. Spectroscopic studies have shown that in proteins, photo-induced disulfide cleavage can be mediated by tryptophan via an electron transfer mechanism[36]. Absorption of the UV photon by tryptophan results in $1\pi \rightarrow \pi^*$ excitation to form the singlet state, 1Trp*. This is followed by relaxation via fluorescence or quenching. One possible quenching pathway

is intersystem crossing to the triplet excited state 3Trp. The triplet state is highly reducing, and is able to donate an electron to the disulfide acceptor leading to the formation of the disulfide radical anion $CSSC^-$ and radical cation $Trp^+$. Electron transfer between tryptophan and disulfides has been observed over distances up to 13 Å[37], suggesting that donor-acceptor pairing between W42 and C41 (under 7 Å in both chains), and between W130 and C108 (under 9 Å in both chains) is possible. This is consistent with the accelerated aggregation of W42E and W130E mutant proteins mimicking UV-induced damage, whereas aggregation of the W68E and W156E mutants are comparable to that of the wild-type[38]. Hence, disulfide cleavage likely proceeds in the most part via tryptophan excitation.

Cleavage of the disulfide bond in $CSSC^-$ produces a thiyl radical on the sulfur closest to the tryptophan; in this case, a Cys radical and a DTT sulfur anion. However, with continuous illumination we observe the formation of sulfenic acid at C41, cysteine at C108 and no obvious changes to the structure of the neighboring tryptophans W42 and W130 in both copies of $HGD_m$ in the asymmetric unit. The resulting fates of the Trp⁺ and Cys⁻ radicals depend on the local environment with numerous possible reaction pathways, including tryptophan oxidation to kynurenine[39], L- to D-isomerisation[40] and backbone cleavage[12]. No evidence of these

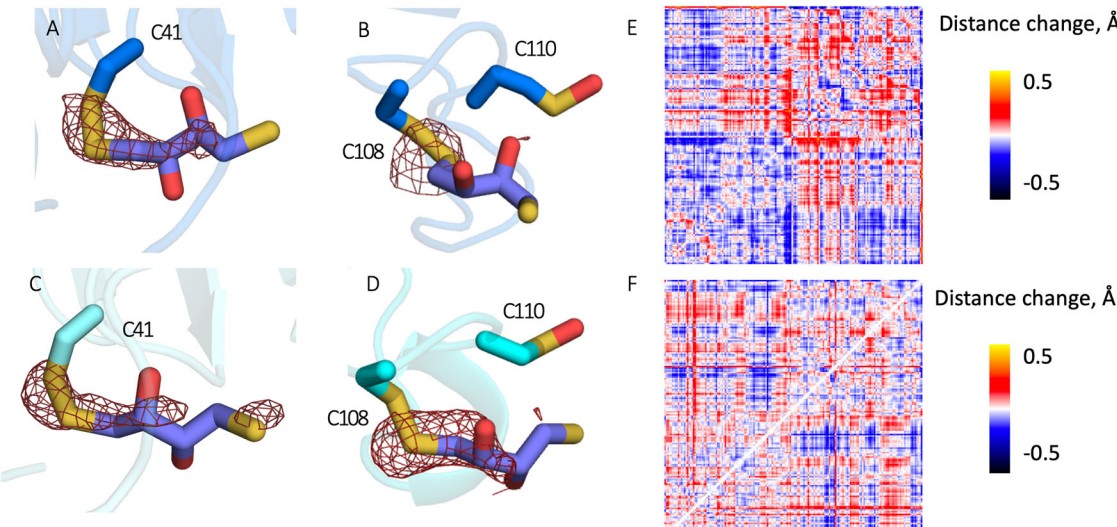

**Fig. 3 | Structural evidence for the UV induced removal of diulfide adducts.** Fo-Fo difference maps contoured at 3 $\sigma$ of the aged dataset from light dataset, centered on (**A**) chain A C41, (**B**) chain A C108 and C110, (**C**) chain X C41 and (**D**) chain X C108 and C110. Atom-to-atom distance difference plots showing effect of global motions exhibited by the first principal component aligned with light structure relative to aged structure for (**E**) chain A and (**F**) chain X.

modifications was observed in the electron density of the light structure nor was it possible to locate the liberated DTT. Consistent with this, cyclisation of DTT would allow backtransfer of an electron to the Trp$^{\cdot+}$ donor, leaving a thiyl radical. The formation of a disulfide bond upon cyclization of DTT acts as a mimic of the disulfide formation in the oxidation of GSH to produce GSSG. The fate of thiyl radicals in the context of radical scavenging has been previously reviewed[41]. We propose that this process contributes to the redox balance within the lens and plays a protective role against aggregation mechanisms resulting from increasing concentration of ROS in the aging lens.

### Insights into a potential repair mechanism

The analysis of atomic coordinate-derived conformational space (RoPE, described in Methods Section 4.6)[30], show that while aging of the crystal produces a large shift in the conformational space, this is partially reversed by UV exposure (Fig. 2A, B). The corresponding interatomic distance plots

also show a reversal of the conformational changes that occurred during aging (Fig. 3E, F) for each chain. However, total reversal is precluded perhaps due to the rate of structural relaxation and oxidation of C41 to sulfenic acid, although a much smaller covalent modification, may continue to influence the conformation.

We therefore tentatively propose a plausible mechanism of crystallin protection based on this structural data, in agreement with prior work[42]. The formation of a disulfide bond, either through S-glutathionylation or dimerization, is potentially reversed by a UV-induced electron transfer redox switch in crystallin. Exposure to UV initiates disulfide cleavage, returning the crystallin to the fresh state. In support of this, we have shown that after UV illumination of aged crystals, HGD$_m$ returns at least in part to the original conformational state of fresh protein. The results suggest that the formation of disulfide bonds initiates a UV-driven redox switch that is able to rescue damaged crystallin byproducts: disulfide cross-linked adducts can be rescued potentially by electron transfer from tryptophan to the disulfide bond, resulting in its cleavage and regeneration of the reducing agent. The presence of DTT-cysteine adducts at C41 and C108 are perhaps counter-intuitive, as C110 is implicated as the major site of protein cross-linking[25]. However, the presence of DTT on C41 and C108 might explain why C110 is the only available cysteine to support dimerization, if the former are naturally protected by GSH in vivo. Additionally, the distance between C110 and the closest tryptophan, W130, is beyond that previously observed for electron transfer[37] and so the proposed mechanism may only be relevant for C41 and C108. The role of UV-mediated disulfide exchange on the delicate redox balance of the lens may go some way to explain the evolutionary conservation of surface cysteines in HGD[43]. These observations also provide evidence to support a putative circadian mechanism of redox homeostasis by the production of GSH in the aqueous humor[8,44].

Coupling of tryptophan UV absorption and electron transfer to cysteine has been observed in other proteins[45–47], and UV-mediated disulfide breaking by this route is likely to far outweigh direct cleavage due to the low molar extinction coefficient of disulfide bonds. Tryptophans within each domain are already known to quench the UV absorption of one another[20]. However, the efficiency of this type of quenching is sensitive to the distance between the tryptophan and disulfide bond moieties, as well as the C-S-S-C dihedral angle. The observed phenomenon of domain contraction when binding to DTT will also have an effect on the quantum yield of electron transfer and must be taken into account in modeling scenarios.

The mechanism we have presented here is putative and falsifiable: mutant structures and independent confirmation of the post-translational

**Table 2 | Calculated pKa with either ProPka or PyPka, and percentage burial of surface cysteine residues C41, C108 C110 on chain X and A**

| Chain - Residue | ProPka-calculated pKa | % surface burial | PyPka-calculated pKa |
|---|---|---|---|
| A - C41 | 11.36 | 66 | 10.54 |
| A - C108 | 10.44 | 40 | 10.59 |
| A - C110 | 9.06 | 0 | 9.55 |
| X - C41 | 11.27 | 63 | 10.69 |
| X - C108 | 10.83 | 78 | 10.23 |
| X - C110 | 9.45 | 22 | 9.41 |

**Table 3 | measured C-S-S-C dihedral angles and maximum absorption wavelength**

| Disulfide bond | Dihedral torsion angle / ° | λ max / nm |
|---|---|---|
| C41-DTT chain X | 85.4 | 260.2 |
| C108-DTT chain X | 43.1 | 342.3 |
| C41-DTT chain A | 36.7 | 362.4 |
| C108-DTT chain A | 12.8 | 447.5 |

modifications of HGD by GSH will be necessary to provide direct evidence for GSH involvement and the role of tryptophan in the proposed mechanism. The structural evidence here also provides no comment on the frequency of this mechanism in vivo. Future investigations on the exact nature of the cleavage pathway in HGD would require ultra-fast determination of structural changes combined with X-ray spectroscopic interrogation of the proposed electron transfer mechanism[34]. While this investigation focuses on the UV disruption of disulfide adducts, other post-translational modifications observed in aged crystallins play an important role. For example, recent structural evidence suggests an interplay between deamidation and the redox state of cysteines in gamma-S crystallin[48]. The results presented represent the first structurally supported model of the interaction between UV, the eye lens proteins, reactive oxygen species, and disulfide adducts.

## Methods

### Expression and Purification

The R36S mutant His-tagged $HGD_m$ gene (CRYGD) subcloned into Nde1 and BamH1 sites of pET-30b(+) was purchased from Genscript. The plasmid was transformed into E. coli BL21 (DE3) using the heat shock method. A single colony of the plasmid-containing protein was inoculated in 50 ml of LB medium containing 50 $\mu$g/ml kanamycin and incubated with shaking (200 rpm) at 37 °C overnight. Aliquots of 1 ml were used to inoculate 1 L of autoinduction media (AIM, Formedium) and incubated with shaking (200 rpm) at 18 °C for 24 hours until $OD_{600}$ was 8. Cells were harvested by centrifugation (20 minutes, 5000 × g at 4 °C) and the pellets were flash frozen in liquid nitrogen and stored at -80 °C. Pellets were then thawed and resuspended in Buffer A (50 mM $NaH_2PO_4$, 500 mM NaCl, 10 mM imidazole, 1 mM PMSF, complete protease tablets EDTA-free, 10 % v/v glycerol, pH 8.0) and lysed under high pressure (40 psi) with continuous flow. Lysates were clarified by centrifugation at 12,000 × g for 30 minutes at 4 °C and the supernatant removed. Ni-NTA beads were equilibrated in Buffer A and added to the cleared supernatant. The supernatant/Ni-NTA mixture was incubated for 2 hours at 4 °C and stirred in order to bind the 6-His-tagged proteins. The beads were washed in Buffer B (50 mM $NaH_2PO_4$, 300 mM NaCl, 20 mM imidazole, pH 8.0) and then packed into a gravity flow column. The column was washed with 10 × bed volumes of Buffer B, followed by 10 × bed volumes of Buffer B with 0.1% v/v Triton X-100 and finally 10 × bed volumes of Buffer B adjusted to 1 M NaCl. Bound proteins were eluted from the Ni-NTA beads by addition of 5 × bed volumes Buffer B with increasing concentrations of imidazole (50–500 mM). The eluted protein was concentrated, and buffer exchanged into 50 mM $NaH_2PO_4$, 300 mM NaCl, 10 % v/v glycerol, pH 8.0 using Vivaspin 5 5000 Da MWCO spin columns (Cytiva). Size exclusion chromatography was used to remove a high molecular weight contaminant and resolve monomer and dimer fractions of $HGD_m$. $HGD_m$ was injected into a Superdex 200 16/600 (Cytiva) using a Superloop (Cytiva) and run at 0.8 ml min$^{-1}$ on an AKTA Pure (Cytiva) in the storage buffer 50 mM $NaH_2PO_4$, 300 mM NaCl, 20 mM DTT, pH 8.0.

### Crystallization

Purified protein in the storage buffer was concentrated from 0.5 mg/ml using Vivaspin 5 5000 MWCO spin columns (Cytiva) centrifuged at 4200 × g over a temperature gradient from 19 to 4 °C for 15 minutes. The protein solution volume was reduced to a quater of the initial value and a microcrystalline (5 - 30 μm) slurry formed spontaneously during centrifugation, crystals were stored at 4 °C.

### Serial data collection - fresh sample

After two weeks, X-ray data were recorded from the crystals using fixed target SSX at I24 Diamond Light Source (DLS, UK)[49]. The crystal slurry was agitated using a vortex mixer (500 rpm) and 100 μl of the slurry was pipetted onto the surface of a micro-patterned silicon wafer chip[49,50]. Gentle vacuum suction was used to draw crystals into the wells of the chip and remove excess buffer. The chips were then sealed between two pieces of 6 μm mylar film to reduce evaporation. Data were collected using the shutterless collection mode. This was performed at room temperature with a tophat beam of size 20 × 20 μm with energy 12.4 keV and flux of 8 × $10^{12}$ ph/s at the sample position. Each crystal was exposed to the X-ray beam attenuated to 40% for 20 ms to record a single snapshot image. The diffraction was recorded on a Pilatus3 6M detector at a distance of 320 mm from the sample.

### Serial data collection—aged and light sample

After incubation of the crystal slurry in the dark at room temperature for 9 months, X-ray data were recorded from crystals using fixed target SSX at the T-REXX endstation at beamline P14 EMBL@PetraIII (Germany)[51]. Crystals were loaded into silicon wafer chips. Data were collected at room temperature using a beam of size 15 × 15 $\mu$m with energy 12.65 keV and flux of 2 × $10^{12}$ ph/s at the sample position. A burst series of 20 images was collected from each crystal with 5 ms exposure per image with 100% beam transmission and the diffraction recorded on an Eiger X 16M detector at a distance of 120 mm from the sample. In order to collect data with exposure to UV, the same sample preparation and beamline parameters were used with the addition of continuous UV-LED (Thorlabs M285L5, 20 mW cm$^{-2}$, 285 nm) illumination of the entire chip over the 20 × 5 ms exposures.

### Data processing

Data were processed using CrystFEL (v0.10.2)[52]. Diffraction peaks were identified using the zaef spot-finding algorithm with the pixel value threshold of 20 and minimum peak gradient of 100, and the signal-to-noise ratio cutoff was 5. The xgandalf algorithm was used to index reflections before three-ringed integration with radii of 3, 4, and 5 pixels. The integrated reflections were then merged and scaled with partialator. Resolution cutoff was determined by choosing the highest resolution shell with a CC* of 0.5. Phases were estimated by molecular replacement with Phaser (CCP4 v.4.7.1)[53,54] using a search model from the high-resolution wild-type structure PDB: 1HK0[23]. Isotropic refinement was then performed using multiple rounds of REFMAC5 (10 cycles) (CCP4 v.4.7.1)[55] and manual adjustments in Coot (v.0.9.8.8)[56]. Comparison of Wilson distributions revealed that no additional scaling of the data was required prior to the calculation of Fo - Fo difference maps using Phenix (v1.20.1)[57] with the amplitude threshold set to 3 $\sigma$ and high-resolution cut-off at 2.1 Å. Difference map peaks were calculated using PEAKMAX (CCP4 v.4.7.1). The r.m.s.d. was calculated by superposition of structures using GESAMT (CCP4 v.4.7.1)[58]. The diffraction weighted dose was calculated using RADDOSE-3D[29] assuming cubic crystals, 20 $\mu$m in size.

### Structural analysis using RoPE

In order to determine if the crystal condition (fresh, aged, light) was correlated with protein structure, diffraction images from each sample were split into 5 equally sized bins evenly distributed across the fixed target chip. As any variation between crystals of the same condition would support the null hypothesis, this was considered a valid approach. The $HGD_m$ dark structure was used as a starting model for refinement for each condition. Refinement in REFMAC5 (10 cycles) was followed by automated stepped refinement in Coot and another identical refinement in REFMAC5 before all 15 split structures were passed through RoPE[30]. Conformational spaces generated from differences in atomic coordinate and torsion angle variation heat maps were calculated as previously described[30]. In addition, a new feature was developed in order to visualize the atomic coordinate-derived changes between conditions: normalized units of the SVD-derived **U** matrix basis vectors were converted to coordinate motions for each atom as previously described for torsion angles[30]. This coordinate motion per atom was added onto corresponding atom positions of the first fresh batch structure and differences in atom-atom pair distances were plotted as a 2D matrix.

### Reporting summary

Further information on research design is available in the Nature Portfolio Reporting Summary linked to this article.

## Data availability
Structure factors and atomic models have been deposited on the Protein Data Bank under the accession codes 8Q3L (R36S fresh HGD, Supplementary Data 1), 8BD0 (R36S aged HGD, Supplementary Data 2) and 8BPI (R36S light HGD, Supplementary Data 3).

## Code availability
The code used for the analysis in this paper is available in commit c46d968 of repository at www.github.com/helenginn/rope.git.

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

## Acknowledgements
The SSX data were collected at beamline I24 at Diamond Light Source (beam time allocation: NT27314-27, Diamond Light Source ltd, United Kingdom) and the T-REXX endstation at beamline P14 operated by EMBL Hamburg at the PETRA III storage ring (beam time allocation: MX471, DESY, Hamburg, Germany). T-REXX is funded by the Bundesministerium für Bildung und Forschung (BMBF), 05K16GU1, 05K19GU1, 05K22GU6. Briony Yorke is funded by the Academy of Medical Sciences Springboard award (SBF006044). Helen Ginn is funded by the Helmholtz Association, grant VH-NG-19-02 (Helmholtz Young Investigator Group). Arwen Pearson is funded by the Federal Excellence Cluster, AIM: advanced imaging of matter (DFG, EXC2056). The authors would like to thank Nils Huse for helpful discussions regarding the photochemistry of disulfide bonds.

## Author contributions
B.A.Y. and A.R.P. designed and initiated the project. J.A.H. and Y.N. produced, characterized and crystallized protein samples. J.A.H., S.H., D.v.S., D.A., R.L.O., A.R.P. and B.A.Y. collected and processed serial crystallographic data. J.A.H., B.A.Y. and H.M.G. analyzed crystallographic data. G.S.B. and Y.N. contributed to analysis of disulfide chemistry. H.M.G. devised and performed the torsion angle analysis. H.M.G. and B.A.Y. generated figures and prepared the manuscript with input from all authors.

## Competing interests
The authors declare no competing interests.
