## [Peer Review File · Communications Chemistry]

Reviewers' comments:

Reviewer #1 (Remarks to the Author):

Hill et al conducted a serial synchrotron crystallography experiment to investigate the impact of oxidative stress on the eye lens protein, human gamma-D crystallin. By applying state-of-the-art SSX techniques to this intriguing subject, this study is anticipated to offer readers valuable insights into various scientific domains. Nevertheless, the manuscript exhibits several experimental gaps and lacks essential information necessary for readers to comprehend the findings. Furthermore, although the manuscript's data analysis and discussions are engaging, the presentation of results, such as figures, is notably limited. I am of the opinion that the manuscript could benefit from enhancement in the following areas:

1. In the case of the structure written by the author, checking the electron density map is essential to determine whether the structural analysis is clear. The structures currently deposited in PDB are in an unreleased state. Therefore, relevant data (pdb & mtz) must be submitted to reviewers to verify the accuracy of model building and electron density maps.

2. "After aging, DTT adduct formation is observed at 100% occupancy at C41 and C108 and oxidation at C110 to

sulfenic acid." How can author prove 100% occupancy? The B-factor value of the ligand in Table 1 is higher than that of protein or water.

3. Table 1: I assume the overall CC1/2 for R36S HGD Aged is a typo. And the highest CC1/2 value for R36S HGD Aged and Light (5ms) is higher than the general standard. What is the resolution cutoff standard? Authors should indicate the highest value for Rsplit value.

4. Information on serial crystallography experiments is lacking. How many indexed images were used to determine the structure? Were the samples exposed to light during data collection? A detailed explanation of the fixed-target scan method is needed.

5. What is the size of the UV-LED illumination in Time-resolve SSX? Is the UV-LED illumination designed to only expose the sample once? Can an UV-LED illumination exposed to a fixed-target affect surrounding samples? Other than that, very detailed information is needed.

6. How can the author store the samples without light for 9 months? Aren't proteins already exposed to light during the production, purification, and crystallization process, and aren't crystals also exposed to light during the process of observing them under a microscope to check their growth or delivering samples at beam time?

7. Figures: Except for Fo-Fo difference maps, the information provided by other structures is very limited. Also, an unusual font is used. I think it would be appropriate to add a figure that can better illustrate the author's argument.

Reviewer #2 (Remarks to the Author):

Review: Communications Chemistry manuscript COMMSCHEM-23-0499-T

Hill et al., "An ultraviolet-driven rescue pathway for oxidative stress to eye lens protein human gamma-D crystallin"

In this manuscript, Hill et al report an interesting finding regarding adduct formation between DTT – a reducing agent commonly used to maintain protein-cysteine thiols in the reduced state – and a cataract-associated mutant of the human lens protein, gammaD-crystallin. They show that microcrystals of the mutant gammaD-crystallin (Arg36Ser or R36S) – upon incubation with 20 mM DTT at pH 8 in the dark for 9 months – leads to the formation of an adduct with DTT. In addition, one Cys residue of the protein is oxidized to sulfenic acid. Using low-dose serial synchrotron crystallography (SSX), they have determined crystal structures of a DTT-bound form of the protein and shown how the bound DTT can be cleaved by UVB radiation at a fixed wavelength (285 nm). The authors also report that UV-light at 285 nm induces lability of the adduct and partially restores the protein structure to its native state. Based on these observations the authors propose that this process could be a mechanism, also in the lens in vivo, by which UV light may actually confer some protection against ultraviolet radiation damage and rescue lens proteins.

Since Cleland's reagent (DTT) is typically not believed to form adducts or mixed disulfides with protein thiols because its cyclic form is kinetically favored, this finding is potentially significant. It might be perceived as novel by researchers unfamiliar with already published work. However, the observation that DTT forms an adduct with protein thiols is not entirely novel, and was first reported by Sheraga's group in RNase A (Li, Rothwarf and Sheraga, J. Am. Chem. Soc. 1998, 120, 2668-2669), and later by another group in aged samples of adenylate kinase (Li, Xian and Pan, 2001, FEBS Letters 507, 169-173). But - the determination of crystal structures showing adduct formation with DTT may be novel. A quick search of the Protein Data Bank by this reviewer did not produce any structures showing bound DTT – hence this

report could very well be the first of this kind, but a more thorough search is warranted. Statements regarding mechanisms in vivo are not plausible and are far-fetched based on the data.

There are several additional concerns and questions that need to be addressed before this work can be considered for publication. These are listed below:

- An important control is missing from their experimental protocols. The authors compare protein samples when “fresh” (after two weeks at 4 degreesC) and after 9 months of aging at room temperature in the dark, both incubated with 20 mM DTT. What about incubating protein alone for a long time without DTT? Others have shown that simple aging of gamma crystallins at pH 7 leads to disulfide crosslink formation. Do those disulfide crosslinks also get cleaved with 289 nm radiation?
- The authors’ conclusions regarding adduct formation would be strengthened if they could show using mass spectrometry that the mass of the protein adduct is consistent with the theoretical/calculated mass. (Conventional x-ray data – even at about 2 Å resolution – sometimes cannot distinguish between an aromatic residue and a chloride ion. This may not be true for SSX – it’s not clear what the resolution of the measurements presented here are).
- The authors state that the DTT-adduct is not fully cleaved at 285 nm and speculate that this may be due to the low quantum yield. Could this not be verified by irradiating with higher flux to possibly increase the concentration of the cleaved product? Dose response studies even at a single wavelength would be revealing. This is particularly important because the cornea absorbs about 75% of UVB radiation and is the main protector of the retina even before the lens takes over.
- If the pKa of C110 (9-9.5) is calculated to be lower than that of C41 and C108, would using pH 7 for these studies prevent sulfenic acid formation? Just about all studies with the crystallins are carried out at physiological pH values around 7.
- Using the R36S mutant for these studies, while convenient for microcrystal formation, is not quite appropriate. Although the two proteins have nearly identical protein fold, their crystal packing is distinct. Thus, it is not clear from their data if HGD and R36S will form a DTT adduct the same way. Moreover, while the His-tag used clearly enables the authors to concentrate the protein to much higher values (20 mg/mL versus less than a mg/mL without the tag), it has not been removed and is still present during SSX measurements. Would removing this tag alter the results? Also important – referring to the cataractogenic R36S mutant as HGD is totally misleading! This is a model system and should be designated as such and the protein should be renamed (for example HGD-m for mutant, or any other name).

- Since DTT is not the natural reductant in the lens, it would make more sense to show if glutathione (the natural reductant which does form mixed disulfides) adducts are reversible by UVB radiation. GSH does not cyclize upon oxidation, so it would be very significant to know whether these two reductants differ in their response, and if so how. More importantly, studies with GSH would be more meaningful in extrapolating to an in vivo mechanism rather than speculation based on the current data with DTT.

- The authors seem to propose that HGD, the most abundant gamma crystallin in the human lens acts as a redox-switch, simply based on their data (the redox-switch idea has also been floated by other researchers on more rigorous biochemical/biophysical data). But the human lens is neither made up of HGD crystals (for obvious reasons), nor is it merely a bag of gammaD-crystallin solution. It has a cellular structure with a mixture of various crystallin types, many containing Cys residues. There are numerous other proteins, including membrane proteins, as well as free Cysteine and Cystine, and high concentrations of GSH and GSSG. Thus, adduct formation and crosslinks of multiple types are possible, and the contribution of a single protein adduct may not be direct. The authors do not seem to recognize this fact.

Other – relatively minor – errors/omissions include:

Abstract: There is no mention of the reducing agent or that the model protein used is a cataractogenic mutant. It reads as though HGD is used for the study.

Main, pg 2: Why present such a detailed chemistry of GSH reactions and products when it is not used in this study? How does this relate to DTT?

Page 6, para 2: “Cyclization of DTT acts as a mimic of GSH dimerization”. How? GSH does not cyclize upon dimerization.

References: Reference no. 19 is incomplete. Also - providing DOI links for the literature cited would be useful, unless the journal format does not require this??

Reviewer #3 (Remarks to the Author):

Hill et al. report serial synchrotron X-ray (SSX) crystal structures of human gamma-D crystallin samples that were (1) freshly purified, (2) aged with exogenous thiols, or (3) aged with exogenous thiols and then UV-irradiated. The “crystal cataract” variant R36S is used in lieu of the WT protein solely to facilitate crystallization under the authors’ preferred conditions. The “aged” samples are incubated for 9 months in the presence of dithiothreitol (DTT). While initially serving as a reducing agent, DTT turns into an oxidizing agent over time as it reacts with dissolved oxygen in the sample. Thanks to their dataset of many crystals imaged with low dose X-rays and averaged, the authors are able to discern for the first time clear electron densities corresponding to two Cys residues in this protein (Cys41 and Cys108) being disulfide-bonded to DTT, while a third Cys residue (Cys110) is observed to be converted to a sulfenic acid residue.

Previous studies using mass spectrometry have established that disulfide bonds and S-glutathionylation occur in aged lens crystallins, but the latter has not been observed directly by crystallography, nor, to my knowledge, has anyone observed a stable sulfenic acid modification on a Cys residue. In fact, many of us have considered a sulfenic acid modification in crystallins to be very unlikely due to the short half-lives of sulfenic acids in vivo – see, e.g., <https://doi.org/10.1016/j.bbagen.2013.05.040>. Therefore, the observation of such a modification persisting apparently for months in human gamma-D crystallin is striking and noteworthy. Gamma-D crystallin was recently shown to be capable of carrying an exceptionally long-lived free radical, as well (<https://doi.org/10.1021/jacs.2c13397>), raising the possibility that this protein is particularly able to stabilize otherwise labile chemical moieties.

Equally noteworthy is the observation of UV-induced disulfide scission in human gamma-D crystallin, and the authors make a plausible argument that UV absorbance by Trp residues catalyzes the subsequent disulfide scission. Exactly this mechanism has already been proposed to function in Cys-rich gamma-crystallins: see <https://doi.org/10.1016/j.exer.2021.108707>. The authors therefore present interesting albeit indirect experimental evidence to support this hypothesis. To more fully support it, two pieces of evidence are missing: (1) evidence from mass spectrometry that UV irradiation eliminates the disulfide-linked adducts, as opposed to, e.g., further damaging the protein in ways that prevent crystallization; and (2) point mutations of Trp residues to non-aromatics such as Ile, demonstrating that the mutants do not efficiently rescue themselves from thiol adducts via UV exposure. These future experiments will be very important to conduct, although they should not be required for the publication of the present manuscript.

Overall, this study is an original and very important contribution to the protein chemistry of lens crystallins and therefore the pathology of age-onset cataract disease. It is methodologically sound, well-written, and contains very insightful discussion of the chemistry of disulfide scission. It does suffer from several inaccurate statements and could be better placed in the context of recent relevant research in

the lens crystallin field; in addition, certain caveats and limitations should be more fully discussed. My comments below are intended to help improve the manuscript presenting this valuable study.

1. The first sentence of the Abstract states that HGD is “the major constituent of the eye lens.” This is not accurate. While HGD is one of the abundant beta-gamma crystallins in the lens, the most abundant human lens proteins are the two alpha-crystallins. The authors should cite proteomic datasets (such as this one: <https://doi.org/10.1167/iops.10-7094>) when making statements about relative protein abundance.

2. The abstract states that “a covalently bound reducing agent” is observed in the SSX structures. This is confusing and not really accurate. As mentioned above, DTT turns into an oxidizing agent during the course of a 9-month incubation, and this is the likely reason it can form covalent adducts with proteins in this study. I suggest that the authors should either specify in the abstract that DTT is the adduct or at least rephrase the claim to something like “thiol adducts at Cys residues are observed.”

3. The first sentence of the main text states that high levels of HGD expression are required for proper viscosity of the lens cytosol. Do the authors have a reference for this claim? I am not aware of evidence that HGD is necessary for modulating lens viscosity.

4. Main text paragraph 2 implies that increasing [ROS] is what depletes glutathione in the aging lens, but this is only part of the story. An important missing piece here is the diffusion barrier that prevents the reservoir of reduced glutathione from being replenished in the aging lens. See the original report at <https://10.1006/exer.1998.0549> and recent reviews, e.g., <https://doi.org/10.1016/j.exer.2016.06.018> and <https://doi.org/10.1016/j.exer.2021.108707>.

5. Paragraph 7 of Results states that W42E and W130E mutations accelerate UV-induced aggregation, but the study in ref. 33 did not UV-irradiate proteins. It did demonstrate that mutational damage mimicking UV-induced damage is sufficient by itself to cause misfolding and aggregation under physiological conditions. However, when combined with evidence in ref. 15 that mutating those Trp residues to Phe promotes UV-induced aggregation, the authors’ interpretation is probably justified.

6. This study is similar in spirit to a recent crystallographic study of lens crystallin disulfide formation upon aging and deamidation: <https://doi.org/10.1016/j.str.2022.03.002>. It would be useful to discuss the conclusions of the present study in the context of that precedent, given that it yielded a different set of observations.

7. The authors propose that their observed DTT adducts are a good model for glutathione adducts that form in vivo. Unfortunately, no experiments with glutathione have been conducted in the present study. It is important to note that glutathione is much bulkier than DTT and is charged. While DTT adduction at partially buried sites (Cys41, Cys108) induced subtle small though widespread conformational shifts, adduction of GSH may perturb the protein much more dramatically. For example, it could lead to disruption of the domain interface in HGD, which is known to be very important in the kinetic stability of the aggregation-prone N-terminal domain (see, e.g., <https://doi.org/10.1016/j.bpj.2019.06.006>).

8. An important caveat is that these structures were obtained from crystals grown at high pH (pH 8.0, per the Methods). This is above the physiological cytoplasmic pH of the lens core fiber cells (pH ~6.8, according to <https://doi.org/10.1085/jgp.98.6.1085>). Since the pKa of Cys is typically ~8, the elevated pH in these experiments is expected to facilitate thiol/disulfide exchange compared to physiological conditions.

9. Lastly, the discussion of Trp-catalyzed scission of disulfides in HGD, while very plausible, is still speculative. As mentioned above, mass spec and mutational evidence would likely be required to establish this more firmly. The authors should discuss this limitation and the need for further experiments to test the proposed mechanism.

Reviewer #1 (Remarks to the Author):

Hill et al conducted a serial synchrotron crystallography experiment to investigate the impact of oxidative stress on the eye lens protein, human gamma-D crystallin. By applying state-of-the-art SSX techniques to this intriguing subject, this study is anticipated to offer readers valuable insights into various scientific domains. Nevertheless, the manuscript exhibits several experimental gaps and lacks essential information necessary for readers to comprehend the findings. Furthermore, although the manuscript's data analysis and discussions are engaging, the presentation of results, such as figures, is notably limited. I am of the opinion that the manuscript could benefit from enhancement in the following areas:

1. In the case of the structure written by the author, checking the electron density map is essential to determine whether the structural analysis is clear. The structures currently deposited in PDB are in an unreleased state. Therefore, relevant data (pdb & mtz) must be submitted to reviewers to verify the accuracy of model building and electron density maps.

We apologise for this error; we agree that PDBs and MTZs should be deposited in a released state before review and have corrected this.

2. "After aging, DTT adduct formation is observed at 100% occupancy at C41 and C108 and oxidation at C110 to sulfenic acid." How can author prove 100% occupancy? The B-factor value of the ligand in Table 1 is higher than that of protein or water.

For the first submission of this manuscript we refined the occupancies in Refmac. However it is certainly true that the B factor differences between the two sulphurs in the disulphide bridge are reasonably variable and we cannot immediately attribute this to increased flexibility of the bound moiety.

So, we established minimum occupancies based on the difference in B factors (A41 of 65.9%, A108 of 64.4%, X41 of 60.1% and X108 of 74.6%), which is the corresponding reduction in peak height if the occupancy was incorrectly and fully replaced by a B factor change. We then tried a number of manual Refmac runs with values in between this minimum and maximum occupancy range. However the Rwork/Rfree did not improve and in fact was best for the 100% occupancy run. Therefore we feel that the data have been as well-described by the model as possible given the drawbacks of occupancy analysis by B factor in light of a single bulk solvent model.

3. Table 1: I assume the overall CC1/2 for R36S HGD Aged is a typo. And the highest CC1/2 value for R36S HGD Aged and Light (5ms) is higher than the general standard. What is the resolution cutoff standard? Authors should indicate the highest value for Rsplit value.

Thank you, the typo is corrected for the overall CC1/2 value. The resolution cutoffs were based on CC as recommended by Karplus & Diederichs. The highest value for Rsplit has been added to the crystallographic statistics and the resolution cutoff standard was added to the methods.*

4. Information on serial crystallography experiments is lacking. How many indexed images were used to determine the structure?

These have been added to the crystallographic statistics table.

Were the samples exposed to light during data collection?

We have added more detail to the methods section to clarify.

A detailed explanation of the fixed-target scan method is needed.

We have clarified the details of the methods in line with reporting standards for serial crystallography. This typically does not involve an extended explanation of fixed-target scanning methods but we have referenced the papers in which this technology was originally developed.

5. What is the size of the UV-LED illumination in Time-resolve SSX? Is the UV-LED illumination designed to only expose the sample once? Can an UV-LED illumination exposed to a fixed-target affect surrounding samples? Other than that, very detailed information is needed.

We have updated the methods section to clarify and removed 'time-resolved', the crystals were exposed to UV during the entire experiment and the data presented represents a UV exposed steady-state.

6. How can the author store the samples without light for 9 months? Aren't proteins already exposed to light during the production, purification, and crystallization process, and aren't crystals also exposed to light during the process of observing them under a microscope to check their growth or delivering samples at beam time?

The protein was exposed to light during the production of protein and initial crystallisation, however during aging the crystals were stored in an opaque box. While, small sample of the crystalline slurry was checked under the microscope, these crystals were not used in the experiment. All crystals were exposed to some ambient light during data collection, however, the UV emission from indoor lighting is negligible.

7. Figures: Except for Fo-Fo difference maps, the information provided by other structures is very limited. Also, an unusual font is used.

We have added another figure to illustrate the adduct formation.

I think it would be appropriate to add a figure that can better illustrate the author's argument.

We are lacking sufficient clarity to understand which argument you are referring to here. However, we have added 2mFo-dFc maps to the main text figure in the course of updating the manuscript.

Reviewer #2 (Remarks to the Author):

Review: Communications Chemistry manuscript COMMSCHEM-23-0499-T

Hill et al., "An ultraviolet-driven rescue pathway for oxidative stress to eye lens protein human gamma-D crystallin"

In this manuscript, Hill et al report an interesting finding regarding adduct formation between DTT – a reducing agent commonly used to maintain protein-cysteine thiols in the reduced state – and a cataract-associated mutant of the human lens protein, gammaD-crystallin. They show that microcrystals of the mutant gammaD-crystallin (Arg36Ser or R36S) – upon incubation with 20 mM DTT at pH 8 in the dark for 9 months – leads to the formation of an adduct with DTT. In addition, one Cys residue of the protein is oxidized to sulfenic acid. Using low-dose serial synchrotron crystallography (SSX), they have determined crystal structures of a DTT-bound form of the protein and shown how the bound DTT can be cleaved by UVB radiation at a fixed wavelength (285 nm). The authors also report that UV-light at 285 nm induces lability of the adduct and partially restores the protein structure to its native state. Based on these observations the authors propose that this process could be a mechanism, also in the lens in vivo, by which UV light may actually confer some protection against ultraviolet radiation damage and rescue lens proteins.

Since Cleland's reagent (DTT) is typically not believed to form adducts or mixed disulfides with protein thiols because its cyclic form is kinetically favored, this finding is potentially significant. It might be perceived as novel by researchers unfamiliar with already published work. However, the observation that DTT forms an adduct with protein thiols is not entirely novel, and was first reported by Sheraga's group in RNase A (Li, Rothwarf and Sheraga, J. Am. Chem. Soc. 1998, 120, 2668-2669), and later by another group in aged samples of adenylate kinase (Li, Xian and Pan, 2001, FEBS Letters 507, 169-173). But - the determination of crystal structures showing adduct formation with DTT may be novel. A quick search of the Protein Data Bank by this reviewer did not produce any structures showing bound DTT – hence this report could very well be the first of this kind, but a more thorough search is warranted.

We don't want to suggest that DTT adduct formation is novel here - in fact we have found another example, 1kzi, which has a DTT-Cys modelled into the electron density. We would suspect some interaction as DTT is frequently required to prevent dimerisation, which would otherwise interfere with crystallisation.

Statements regarding mechanisms in vivo are not plausible and are far-fetched based on the data.

Generation of falsifiable hypotheses focuses the interest and relevance of the presented work. We do not want to suggest that we have sufficient evidence to set any in vivo mechanism in stone! We have proposed a (plausible) mechanism, rather than provided statements about in vivo behaviour. However, in light of this comment, we have altered the language to make this especially clear. We have also expanded the discussion in the conclusions to make it clear we cannot draw conclusions in vivo.

There are several additional concerns and questions that need to be addressed before this work can be considered for publication. These are listed below:

- An important control is missing from their experimental protocols. The authors compare protein samples when “fresh” (after two weeks at 4 degreesC) and after 9 months of aging at room temperature in the dark, both incubated with 20 mM DTT. What about incubating protein alone for a long time without DTT? Others have shown that simple aging of gamma crystallins at pH 7 leads to disulfide crosslink formation. Do those disulfide crosslinks also get cleaved with 289 nm radiation?

This experiment would not actually be a control for any of the other experiments that we have carried out - this is a different experiment. Without DTT, the protein would fail to crystallise in the first place and so any experiments that could be performed would need to be done in solution. We would not expect the dimer to be cleavable under these conditions because the dimer is expected to be generated across the C110 residues and this disulphide bond is over 15 Å away from Trp130, excluding the tryptophan-accelerated UV-lability. We have clarified this in the text.

- The authors’ conclusions regarding adduct formation would be strengthened if they could show using mass spectrometry that the mass of the protein adduct is consistent with the theoretical/calculated mass. (Conventional x-ray data – even at about 2 Å resolution – sometimes cannot distinguish between an aromatic residue and a chloride ion. This may not be true for SSX – it’s not clear what the resolution of the measurements presented here are).

We apologise for the oversight that we did not contain the weighted density maps for the DTT. The Fo-Fo maps that we have included in the original draft are considered to be a stronger evidence than 2mFo-dFc maps, however the true shape of the molecule is generally lost. We have therefore updated the paper to include the 2mFo-dFc maps which clearly indicate DTT is an excellent fit to the electron density. The resolution of all collected datasets are given in Table 1.

- The authors state that the DTT-adduct is not fully cleaved at 285 nm and speculate that this may be due to the low quantum yield. Could this not be verified by irradiating with higher flux to possibly increase the concentration of the cleaved product? Dose response studies even at a single wavelength would be revealing. This is particularly important because the cornea absorbs about 75% of UVB radiation and is the main protector of the retina even before the lens takes over.

As the crystals were constantly illuminated with UV we believe that the reference to quantum yield is not actually relevant. So we thank the reviewer for highlighting this point and have altered the manuscript accordingly.

- If the pKa of C110 (9-9.5) is calculated to be lower than that of C41 and C108, would using pH 7 for these studies prevent sulfenic acid formation? Just about all studies with the crystallins are carried out at physiological pH values around 7.

Whilst using pH 7 for these studies would be preferable, we were unable to obtain crystals at this pH. We have addressed the difference in pH levels explicitly in a reply to point 8 to reviewer #3.

- Using the R36S mutant for these studies, while convenient for microcrystal formation, is not quite appropriate. Although the two proteins have nearly identical protein fold, their crystal packing is distinct. Thus, it is not clear from their data if HGD and R36S will form a DTT adduct the same way.

Although the wild-type and R36S mutant have different crystal forms, it is worth noting that neither of these crystal forms are physiologically relevant as wild-type crystallin does not form crystals in the lens. This means neither packing arrangement is going to be a completely accurate representation of the crystallin environment in vivo. The DTT adduct formation is likely to depend on the pKa of the surface individual cysteine residues, these are in good agreement for R37S and the wildtype protein (PDB: 1HK0) C41 (10.49), C108 (10.52), C110 (9.78).

Moreover, while the His-tag used clearly enables the authors to concentrate the protein to much higher values (20 mg/mL versus less than a mg/mL without the tag), it has not been removed and is still present during SSX measurements. Would removing this tag alter the results? Also important – referring to the cataractogenic R36S mutant as HGD is totally misleading! This is a model system and should be designated as such and the protein should be renamed (for example HGD-m for mutant, or any other name).

We have altered the text throughout to ensure that it is clear the mutant protein was used in these studies. While the His-tag is present in the protein we are unable to comment on whether it would alter the results, however the His-tag is disordered and not visible in the crystal structure, and therefore we assume it is not involved in the reported observations. We do not want to comment on the solubility of the his-tagged protein as crystallisation occurs spontaneously during the concentration process - we have updated the manuscript to reflect this.

- Since DTT is not the natural reductant in the lens, it would make more sense to show if glutathione (the natural reductant which does form mixed disulfides) adducts are reversible by UVB radiation. GSH does not cyclize upon oxidation, so it would be very significant to know whether these two reductants differ in their response, and if so how. More importantly, studies with GSH would be more meaningful in extrapolating to an in vivo mechanism rather than speculation based on the current data with DTT.

We agree and are endeavoring to produce crystals grown in high concentrations of GSH.

- The authors seem to propose that HGD, the most abundant gamma crystallin in the human lens acts as a redox-switch, simply based on their data (the redox-switch idea has also been floated by other

researchers on more rigorous biochemical/biophysical data). But the human lens is neither made up of HGD crystals (for obvious reasons), nor is it merely a bag of gammaD-crystallin solution. It has a cellular structure with a mixture of various crystallin types, many containing Cys residues. There are numerous other proteins, including membrane proteins, as well as free Cysteine and Cystine, and high concentrations of GSH and GSSG. Thus, adduct formation and crosslinks of multiple types are possible, and the contribution of a single protein adduct may not be direct. The authors do not seem to recognize this fact.

We have updated the introduction to more accurately describe the contents of the lens. Our research presents structural evidence for a protein adduct. Structural biology is an inherently reductionist approach and must be taken in the context of other biophysical data. We have used more measured comments to refer to the putative hypothesis in vivo as mentioned above.

Other – relatively minor – errors/omissions include:

Abstract: There is no mention of the reducing agent or that the model protein used is a cataractogenic mutant. It reads as though HGD is used for the study.

We have altered the abstract to make this clear.

Main, pg 2: Why present such a detailed chemistry of GSH reactions and products when it is not used in this study? How does this relate to DTT?

We appreciate that the text goes into an unnecessary level of detail about GSH chemistry and we have scaled this back in the introduction.

Page 6, para 2: “Cyclization of DTT acts as a mimic of GSH dimerization”. How? GSH does not cyclize upon dimerization.

We have altered the text to clarify that the formation of a disulfide bond in DTT upon oxidation is equivalent to the disulfide bond formation resulting in the oxidation of GSH to GSSG.

References: Reference no. 19 is incomplete. Also - providing DOI links for the literature cited would be useful, unless the journal format does not require this??

We have fixed reference 19.

Reviewer #3 (Remarks to the Author):

Hill et al. report serial synchrotron X-ray (SSX) crystal structures of human gamma-D crystallin samples that were (1) freshly purified, (2) aged with exogenous thiols, or (3) aged with exogenous thiols and then UV-irradiated. The “crystal cataract” variant R36S is used in lieu of the WT protein solely to facilitate crystallization under the authors’ preferred conditions. The “aged” samples are incubated for 9 months in the presence of dithiothreitol (DTT). While initially serving as a reducing agent, DTT turns into an oxidizing agent over time as it reacts with dissolved oxygen in the sample. Thanks to their dataset of many crystals imaged with low dose X-rays and averaged, the authors are able to discern for the first time clear electron densities corresponding to two Cys residues in this protein (Cys41 and Cys108) being disulfide-bonded to DTT, while a third Cys residue (Cys110) is observed to be converted to a sulfenic acid residue.

Previous studies using mass spectrometry have established that disulfide bonds and S-glutathionylation occur in aged lens crystallins, but the latter has not been observed directly by crystallography, nor, to my knowledge, has anyone observed a stable sulfenic acid modification on a Cys residue. In fact, many of us have considered a sulfenic acid modification in crystallins to be very unlikely due to the short half-lives of sulfenic acids in vivo – see, e.g., (<https://doi.org/10.1016/j.bbagen.2013.05.040>) Therefore, the observation of such a modification persisting apparently for months in human gamma-D crystallin is striking and noteworthy. Gamma-D crystallin was recently shown to be capable of carrying an exceptionally long-lived free radical, as well (<https://doi.org/10.1021/jacs.2c13397>), raising the possibility that this protein is particularly able to stabilize otherwise labile chemical moieties.

This is an interesting point. We are a bit hesitant to draw any further conclusions from our data about this, but it certainly warrants attention.

Equally noteworthy is the observation of UV-induced disulfide scission in human gamma-D crystallin, and the authors make a plausible argument that UV absorbance by Trp residues catalyzes the subsequent disulfide scission. Exactly this mechanism has already been proposed to function in Cys-rich gamma-crystallins: see <https://doi.org/10.1016/j.exer.2021.108707>. The authors therefore present interesting albeit indirect experimental evidence to support this hypothesis.

This is clearly a very relevant paper and we have now cited it - very helpful context, thank you!

To more fully support it, two pieces of evidence are missing: (1) evidence from mass spectrometry that UV irradiation eliminates the disulfide-linked adducts, as opposed to, e.g., further damaging the protein in ways that prevent crystallization; and (2) point mutations of Trp residues to non-aromatics such as Ile, demonstrating that the mutants do not efficiently rescue themselves from thiol adducts via UV exposure. These future experiments will be very important to conduct, although they should not be required for the publication of the present manuscript.

We agree with the referee’s thoughts on future experimental direction.

Overall, this study is an original and very important contribution to the protein chemistry of lens crystallins and therefore the pathology of age-onset cataract disease. It is methodologically sound, well-written, and contains very insightful discussion of the chemistry of disulfide scission. It does suffer from several inaccurate statements and could be better placed in the context of recent relevant research in the lens crystallin field; in addition, certain caveats and limitations should be more fully discussed. My comments below are intended to help improve the manuscript presenting this valuable study.

We thank the referee for these rewarding comments!

1. The first sentence of the Abstract states that HGD is “the major constituent of the eye lens.” This is not accurate. While HGD is one of the abundant beta-gamma crystallins in the lens, the most abundant human lens proteins are the two alpha-crystallins. The authors should cite proteomic datasets (such as this one: <https://doi.org/10.1167/iovs.10-7094>) when making statements about relative protein abundance.

The introduction has been expanded and the text has been altered to correct this statement and included the citation. Thank you for pointing out this error.

2. The abstract states that “a covalently bound reducing agent” is observed in the SSX structures. This is confusing and not really accurate. As mentioned above, DTT turns into an oxidizing agent during the course of a 9-month incubation, and this is the likely reason it can form covalent adducts with proteins in this study. I suggest that the authors should either specify in the abstract that DTT is the adduct or at least rephrase the claim to something like “thiol adducts at Cys residues are observed.”

Thank you for catching this. We have altered the text.

3. The first sentence of the main text states that high levels of HGD expression are required for proper viscosity of the lens cytosol. Do the authors have a reference for this claim? I am not aware of evidence that HGD is necessary for modulating lens viscosity.

We agree that this is poorly worded: the optical density of HGD is what it is, and if it were any different then the lens would have evolved to be a different shape to accommodate that. We have edited the text to remove this concept of “requirement”.

4. Main text paragraph 2 implies that increasing [ROS] is what depletes glutathione in the aging lens, but this is only part of the story. An important missing piece here is the diffusion barrier that prevents the reservoir of reduced glutathione from being replenished in the aging lens. See the original report at <https://10.1006/exer.1998.0549> and recent reviews, e.g., <https://doi.org/10.1016/j.exer.2016.06.018> and <https://doi.org/10.1016/j.exer.2021.108707>.

We thank the referee for these references and appreciate their inclusion in the manuscript.

5. Paragraph 7 of Results states that W42E and W130E mutations accelerate UV-induced aggregation, but the study in ref. 33 did not UV-irradiate proteins. It did demonstrate that mutational damage mimicking UV-induced damage is sufficient by itself to cause misfolding and aggregation under physiological conditions. However, when combined with evidence in ref. 15 that mutating those Trp residues to Phe promotes UV-induced aggregation, the authors' interpretation is probably justified.

We agree that providing similar experimental evidence for tryptophan mutations would provide optimum justification here (out of the scope of this particular paper), but that the tryptophan mutation studies to-date provide sufficient precedent.

6. This study is similar in spirit to a recent crystallographic study of lens crystallin disulfide formation upon aging and deamidation: <https://doi.org/10.1016/j.str.2022.03.002>. It would be useful to discuss the conclusions of the present study in the context of that precedent, given that it yielded a different set of observations.

Thank you, we have added this discussion to the paper.

7. The authors propose that their observed DTT adducts are a good model for glutathione adducts that form in vivo. Unfortunately, no experiments with glutathione have been conducted in the present study. It is important to note that glutathione is much bulkier than DTT and is charged. While DTT adduction at partially buried sites (Cys41, Cys108) induced subtle small though widespread conformational shifts, adduction of GSH may perturb the protein much more dramatically. For example, it could lead to disruption of the domain interface in HGD, which is known to be very important in the kinetic stability of the aggregation-prone N-terminal domain (see, e.g., <https://doi.org/10.1016/j.bpj.2019.06.006>).

This is an important caveat of the paper and unfortunately the use of GSH was not compatible with crystallisation required to perform these structural studies. We have discussed the caveats of the proposed mechanism in reference to the in vivo behaviour.

8. An important caveat is that these structures were obtained from crystals grown at high pH (pH 8.0, per the Methods). This is above the physiological cytoplasmic pH of the lens core fiber cells (pH ~6.8, according to <https://doi.org/10.1085/jgp.98.6.1085>). Since the pKa of Cys is typically ~8, the elevated pH in these experiments is expected to facilitate thiol/disulfide exchange compared to physiological conditions.

This is a fair point and we have made a point in the main text about the difference in pH. The pH of the lens is still within a range which comfortably supports disulfide exchange, but it is true that exchange will be theoretically faster at higher pH.

9. Lastly, the discussion of Trp-catalyzed scission of disulfides in HGD, while very plausible, is still speculative. As mentioned above, mass spec and mutational evidence would likely be required to

establish this more firmly. The authors should discuss this limitation and the need for further experiments to test the proposed mechanism.

Thank you: we have modified the text to make it clear that our proposed mechanism is putative and subject to further enquiry, both where it is first introduced and also in the conclusion, including a discussion of the limitations of the data presented.

REVIEWERS' COMMENTS:

Reviewer #2 (Remarks to the Author):

In this revised version, the authors have, for the most part addressed the main concerns raised by this reviewer. The manuscript is sufficiently improved and reads much better than the original. It makes an interesting contribution to the field as a worthwhile model system. I recommend publication.

Please fix some typos and omissions:1) Page 3, change "UV-incuded" to "UV-induced"

2) Page 3, in the paragraph starting with "In order to...", add the reference to the original R36S paper by Knoch et al.,

Human Molecular Genetics, Volume 9, Issue 12, 22 July 2000, Pages 1779–1786,
<https://doi.org/10.1093/hmg/9.12.1779>

3) Page 7, Line 6 from the top, change "towlocate" to "to locate"

Additional comments:

While this statement in the authors' rebuttal report is not relevant here because it does not appear in the text of the manuscript, I should point out that the crystal form of the R36S mutant IS INDEED physiologically relevant (this is what leads to light scattering)..Obviously, the wild-type should not - and thankfully, does not form crystals in the lens.

Authors' rebuttal: "Although the wild-type and R36S mutant have different crystal forms, it is worth noting that

neither of these crystal forms are physiologically relevant as wild-type crystallin does not form crystals in the lens".

Reviewer #3 (Remarks to the Author):

The authors have appropriately addressed my concerns, except two likely oversights noted below:

Page 6: “acceleration of UV-induced aggregation in W42E and W130E mutant proteins” should say “accelerated aggregation of W42E and W130E mutant proteins mimicking UV-induced damage”

The newly included Ref. 47 probably meant to cite this paper instead (by the same authors):
<https://doi.org/10.1016/j.str.2022.03.002>

Reviewer #4 (Remarks to the Author):

The authors represent an interesting result: HGD can form adducts with DTT at multiple cysteines. These adducts are reversed upon UV irradiation. The authors speculate that this UV-driven reversal represents a protective mechanism. Reviewer comments centered on the adequacy of presentation and provision of materials, and on relevance to the functional in vivo context.

Most of the response to reviewer #1 is adequate. I do concur with the reviewer that the presentation of results is limited. I observe that 8Q3L is still held, and that 8BD0 and 8BPI are in inconsistent settings making a direct visual comparison and calculation of difference maps hard. It would have been strongly preferable to include the isomorphous difference maps, but in this case I was able to reproduce the difference map calculation relatively easily. I consider the remaining comments by reviewer #1 satisfactorily addressed.

With respect to the response to reviewer #2, I note

- I agree with the authors that a no-DTT condition is hardly a control for anything. I do note that that DTT after 2 weeks in water at 4 °C is “fresh” only in the sense that adduct formation has not happened yet.

- I agree that although mass spectrometry evidence for the adduct would be nice, I cannot imagine another source for the adduct than DTT.

With respect to the response to reviewer #3, I agree that point mutation of relevant Trp residues would be highly desirable, as it would make the proposed mechanism more convincing (or disprove it!). I concur with the reviewer and the authors that this can be reserved for future work.

With respect to my own query about Figure 3: I thank the authors for their correction—this is light – aged. I infer from the color that this map is contoured at -3 sigma and that we are seeing the disappearance of the covalent adduct. I would like to urge the authors to also provide (a version of the figure with) positive difference density at $+3$ sigma. It is common practice to include both. The same applies to Figure 1.

Further minor comments:

- The sentence “show a photo-induced electron transfer mechanism” in the Significance statement is too strong claim—they show that such a mechanism must exist, but do not disentangle the mechanism itself (e.g. using Trp point mutants). The abstract, in contrast, is worded correctly.
- As far as I can tell from the data, the UV exposure does not reverse to oxidation of Cys 110.

Overall, the described methods are technically sound and the presented results convincing (in a technical sense). I did not have access to the original submission, but it appears that the authors adequately revised the manuscript to give a careful sense of the potential relevance of these findings to the physiological context. I recommend this work for publications following minor revisions.

Reviewer #2 (Remarks to the Author):

In this revised version, the authors have, for the most part addressed the main concerns raised by this reviewer. The manuscript is sufficiently improved and reads much better than the original. It makes an interesting contribution to the field as a worthwhile model system. I recommend publication.

Please fix some typos and omissions:1) Page 3, change "UV-incuded" to "UV-induced"

Done

2) Page 3, in the paragraph starting with "In order to...", add the reference to the original R36S paper by Kmoch et al.,

Human Molecular Genetics, Volume 9, Issue 12, 22 July 2000, Pages 1779–1786,

<https://doi.org/10.1093/hmg/9.12.1779>

3) Page 7, Line 6 from the top, change "towlocate" to "to locate"

Done

Additional comments:

While this statement in the authors' rebuttal report is not relevant here because it does not appear in the text of the manuscript, I should point out that the crystal form of the R36S mutant IS INDEED physiologically relevant (this is what leads to light scattering)..Obviously, the wild-type should not - and thankfully, does not form crystals in the lens.

Authors' rebuttal: "Although the wild-type and R36S mutant have different crystal forms, it is worth noting that

neither of these crystal forms are physiologically relevant as wild-type crystallin does not form crystals in the lens".

Reviewer #3 (Remarks to the Author):

The authors have appropriately addressed my concerns, except two likely oversights noted below:

Page 6: "acceleration of UV-induced aggregation in W42E and W130E mutant proteins" should say "accelerated aggregation of W42E and W130E mutant proteins mimicking UV-induced damage"

The newly included Ref. 47 probably meant to cite this paper instead (by the same authors):

<https://doi.org/10.1016/j.str.2022.03.002>

Thank you, corrected

Reviewer #4 (Remarks to the Author):

The authors represent an interesting result: HGD can form adducts with DTT at multiple cysteines. These adducts are reversed upon UV irradiation. The authors speculate that this UV-driven reversal represents a protective mechanism. Reviewer comments centered on the adequacy of presentation and provision of materials, and on relevance to the functional in vivo context.

Most of the response to reviewer #1 is adequate. I do concur with the reviewer that the presentation of results is limited. I observe that 8Q3L is still held, and that 8BD0 and 8BPI are in inconsistent settings making a direct visual comparison and calculation of difference maps hard. It would have been strongly preferable to include the isomorphous difference maps, but in this case I was able to reproduce the difference map calculation relatively easily. I consider the remaining comments by reviewer #1 satisfactorily addressed.

With respect to the response to reviewer #2, I note

- I agree with the authors that a no-DTT condition is hardly a control for anything. I do note that that DTT after 2 weeks in water at 4 °C is “fresh” only in the sense that adduct formation has not happened yet.
- I agree that although mass spectrometry evidence for the adduct would be nice, I cannot imagine another source for the adduct than DTT.

With respect to the response to reviewer #3, I agree that point mutation of relevant Trp residues would be highly desirable, as it would make the proposed mechanism more convincing (or disprove it!). I concur with the reviewer and the authors that this can be reserved for future work.

With respect to my own query about Figure 3: I thank the authors for their correction—this is light – aged. I infer from the color that this map is contoured at -3 sigma and that we are seeing the disappearance of the covalent adduct. I would like to urge the authors to also provide (a version of the figure with) positive difference density at $+3$ sigma. It is common practice to include both. The same applies to Figure 1.

We have fixed the figures to include +/- 3 sigma. Thank you for noticing this.

Further minor comments:

- The sentence “show a photo-induced electron transfer mechanism” in the Significance statement is too strong claim—they show that such a mechanism must exist, but do not disentangle the mechanism itself (e.g. using Trp point mutants). The abstract, in contrast, is worded correctly.

Changed to “we demonstrate the possibility of a photoinduced electron transfer mechanism”

- As far as I can tell from the data, the UV exposure does not reverse to oxidation of Cys 110.

Overall, the described methods are technically sound and the presented results convincing (in a technical sense). I did not have access to the original submission, but it appears that the authors adequately revised the manuscript to give a careful sense of the potential relevance of these findings to the physiological context. I recommend this work for publications following minor revisions.